# Biomechanical Analysis of the Coordinated Movements of the Therapist’s Hands and Feet during Lumbopelvic Manipulation: A Preliminary Study

**DOI:** 10.3390/healthcare11233023

**Published:** 2023-11-23

**Authors:** Jejeong Lee, Yongwoo Lee

**Affiliations:** 1Department of Physical Therapy, Graduate School, Sahmyook University, Seoul 01795, Republic of Korea; dlwpwjd@gmail.com; 2Department of Physical Therapy, College of Health and Welfare, Sahmyook University, Seoul 01795, Republic of Korea

**Keywords:** spinal manipulation, biomechanics, manual therapy, patient safety, force coordination

## Abstract

Spinal manipulation (SM) is a common manual therapy technique; however, there is limited knowledge regarding the coordination of hand and foot forces during SM. This study investigated the biomechanics of force transmission and generation in the hands and feet of a single therapist who performed pelvic SM on 45 healthy subjects. Two force plates were used to measure the ground reaction forces (GRF) from the feet, and one controller was used to measure the contact hand forces (CHF). The results showed that foot force preceded hand force and that the foot and hand exhibited opposing patterns of force variation. The CHF peak was positively correlated with the CH preload maximum and minimum forces and negatively correlated with the GRF run-down. These findings suggested that the therapist used a coordinated strategy of avoiding weight support with the feet and supporting the weight with the hands to amplify the thrust force. This study provides new insights into the biomechanics of SM and has implications for teaching, motor learning, and safety.

## 1. Introduction

Spinal manipulation (SM) is a therapeutic modality used by various healthcare providers such as physical therapists, osteopaths, and chiropractors. It is often described as a high-velocity low-amplitude force procedure [1]. SM is a complex bimanual motor skill that involves multiple limb coordination and postural control. Clinicians must learn to manage SM safely and effectively [2]. They must partially support their body weight, perform timely weight shifts from the lower extremities to the upper extremities, and transfer force to the patients. SM in the lumbar and pelvic regions is a particularly complex process that requires controlling the patient’s posture while simultaneously shifting the therapist’s weight and muscle effort in an asymmetrical position. In clinical settings, treatment modifications are necessary at every moment during the force transfer process to respond to the patient’s condition [3].

Most studies on SM have been conducted as randomized controlled trials to determine its clinical effectiveness in patients with back and neck pain [4]. SM is currently recommended as a conservative nonsurgical treatment for lower back pain by most clinical guidelines [5,6,7]. Many techniques can be used to achieve this clinical effect; however, SM requires the application of carefully directed force to a specific location [8]. In other words, therapists must understand not only the clinical effects of SM but also the biomechanical differences and quantification to safely apply it to patients. Despite this need, research on the biomechanical mechanisms of SM is still lacking [9].

Quantitative research on SM has been hard to standardize due to the difficulty of measurement and the great differences displayed depending on the location of the measurement [10]. A few studies have reported that the ranges of the preload and maximum thrust during SM can differ by up to 8–10 times [11,12,13]. If the forces generated in the contact hand (CH) during SM are listed in chronological order, the first stage is the maximum preload stage (facilitates thrust by moving the joint to the limit of the physiological range and is characterized by an increase in force through localizing the target joint), the second stage is the minimum preload stage (characterized by a decrease in the preload just before thrust), and the third is the peak stage (the section where the maximum force is applied to the target joint). Many therapists agree that force is generated by the legs and transmitted to the hands [14]. Although SM is a multi-coordinated movement technique between the therapist’s hands and feet, the hands and feet have been studied separately, and most previous studies have been conducted only on the CH [9,15,16,17]. Derian et al. [18] reported on the force generated from the therapist’s feet during SM; however, the CH force (CHF) was not studied. Descarreaux et al. [17] studied hands and feet simultaneously; however, it was a mannequin experiment that only confirmed the delay time of some sections.

In particular, SM applications of the lumbar spine and pelvic girdle often place the patient in a side-lying position and require timely shifting of the partially supported body weight as forces are transferred from the therapist’s feet to the CH and involve multi-limb coordination and postural control [3,14]. Compared with other applications, it requires considerable motor learning and training because it generates force with complex coordination movements. However, while there are descriptions of the CH and body position of the therapist applying the preload, the biomechanics of gaining momentum and amplifying and transmitting force are not fully understood, despite their obvious importance to the force generated during SM [10,14,19]. Therefore, this study aims to characterize the temporal coordination of force generation and transmission during SM using a biomechanical analysis that simultaneously measures the ground reaction force (GRF) and CHF at the therapist’s feet and hands, respectively.

## 2. Materials and Methods

### 2.1. Study Design and Participants

The type of this study is a biomechanical observational study. This study aimed to observe and analyze the biomechanics of force transmission and generation during spinal manipulation performed by a single therapist on healthy subjects.

For this study, healthy subjects were recruited using an online bulletin board. The inclusion criteria were as follows: (1) experience of receiving SM at the sacroiliac joint (SIJ); (2) no discomfort, pain, or neurological symptoms after previous treatment; and (3) willingness to participate in the study. A total of 45 participants were recruited. The exclusion criteria for the study participants were as follows: (1) history of trauma or surgery of the spine, pelvis, and hip joints; (2) pain in the lumbosacral area over the past six months; (3) disc bulge, protrusion, or neurological symptoms such as radiating pain in the legs; (4) spondylolysis or spondylolisthesis; and (5) severe osteoporosis. Two force plates (K-force Plates, KINVENT, Montpellier, France, 2022) were used to measure the force and time of the feet, and a muscle controller (K-force muscle controller, KINVENT, Montpellier, France, 2022) was used to measure the force and time of the CH during SM.

The sample size was calculated through a pilot study. Through a pilot study, we examined the factors influencing specific patterns of hand and foot movements and their impact on the peak values of the CHF. In the pilot study, we observed a Pearson correlation of R = 0.637 between the CHF PreMax and CHF Peak, as well as a correlation of R = 0.42 between the GRF’s run-down and CHF PreMax. Using G*Power (G-Power software 3.1.2; Franz Faul, University of Kiel, Kiel, Germany), we determined the required sample size for the correlation analysis. Setting the minimum correlation value at 0.42, alpha level at 0.05, and power at 0.8, we obtained a total sample size of 42 individuals. To account for the possibility of not achieving the experimental criterion of cavitation, we added 10%, resulting in the recruitment of 46 participants.

The purpose, process, and expected effects of the experiment were explained to all participants prior to their participation. Informed consent was obtained from all participants, and the study was approved by the Sahmyook University Research Ethics Committee (IRB approval number: SYU 2022-04-008-004).

### 2.2. Research Measurements

The therapist placed the subject in the SM preparatory posture, adjusted their posture, moved into a position where force could be effectively generated, and marked the positions of the left and right feet on the floor. Each force plate was placed at the indicated location. The location of the SIJ was confirmed by palpating the posterior superior iliac spine. The muscle controller was placed on the therapist’s palm who quickly touched it to the force plate of the foot to temporally synchronize the measurements of the CH and foot. The therapist stood on the force plates (Figure 1a) and applied SM to the pelvis by contacting the subject’s SIJ with a controller attached to their palm (Figure 1b). The subject was asked to lie in a side-lying position on the table and close to the therapist. The subjects’ arms were instructed to cross each other. The therapist straightened the leg beneath the subject and had the ankle of the leg positioned above the table rest behind the knee of the lower leg (Figure 2). The therapist held the patient’s arms crossed at the sternum with both hands in the cephalic direction. With the caudal hand, the therapist touched the posterior superior iliac spine of the pelvis with the entire palm, focusing more on the pisiform area. When the restrictive barrier reached the point of tension by pulling the caudal hand located on the subject’s SIJ, SM was performed using the anterior push method, which creates a high-velocity low-amplitude thrust using a body drop. The height of the table was adjusted to match the high point of the knees where the therapist could maintain balance while shifting his weight (Figure 3A,B) [20,21,22]. The force and time of the GR and CH during SM were sampled at 75 Hz, stored, and analyzed using K-force Pro (Ver2022, KINVENT, France). SM thrust was completed in 1 s, but force and time data were collected for a total of 35 s considering temporal synchronization with the CH, subject position control, and body shape control time.

### 2.3. Statistical Analysis

The time of the hands and feet was synchronized, and the preload minimum at which the CH thrust occurred was set at 0 s [16]. The measured force and time data were saved with the K-force Pro (K-force Pro, KINVENT, France, 2022) program. All data statistical analyses were performed using the SPSS (IBM SPSS statistics version 29.0 for Mac, IBM Co., Armonk, NY, USA, 2022) program. To generate force during SM, the coordination relationship and characteristics of the force and time of the GR generated between the therapist’s feet and the ground and those of the CH generated between the therapist’s hands and the skin contact surface were analyzed. To this end, a one-sample *t*-test and Pearson’s correlation analysis were performed.

A one-sample *t*-test was used to analyze the force and time applied to the participants. Pearson’s correlation analysis was used to analyze the correlation between the forces generated over time in the therapist’s hands and feet. The level of statistical significance was set at *p* < 0.05.

## 3. Results

A total of 45 healthy adult men and women were recruited (26 men, 19 women). The data from one participant were excluded as the cavitation sound, which is the standard for successful experimentation, could not be obtained. Subsequently, the results from 44 people were used (Table 1). The test subjects’ height ranged from 158 to 182 cm, with an average of 170.77 ± 6.58 cm. Their age ranged from 24 to 37 years, with an average of 28.82 ± 3.28, and their weight ranged from 47 to 110 kg, with an average of 70.32 ± 14.59 kg (Table 1).

The force and time of the GR generated between the therapist’s feet and the ground during SM were 225.28 ± 55.47 N, −0.34 ± 0.06 S in the run-up. The peak was found to be 617.48 ± 97.41 N, −0.15 ± 0.04 S; the run-down was found to be 139.98 ± 64.20 N, 0.11 ± 0.04 S.

The force and time generated in the CH between the therapist’s hands and the skin contact surface were found to be 288.26 ± 43.89 N, −0.29 ± 0.06 S in the PreMax and 118.67 ± 26.51 N in the preload minimum (PreMin). The peak was 544.08 ± 64.35 N, 0.21 ± 0.03 S (Table 2, Figure 4).

The duration of the GRF in SM was found to be 0.19 ± 0.05 s and 0.26 ± 0.04 s from the run-up to the peak and from the peak to the run-down, respectively. The total time from the run-up to the run-down was 0.46 ± 0.06 s.

The duration of the CHF was 0.29 ± 0.06 s from the PreMax to the PreMin, 0.21 ± 0.03 s from PreMin to peak, and the total time from the PreMax to the peak was 0.50 ± 0.05 s (Table 3, Figure 4).

Over time during SM, the run-up of the GRF and the PreMax CHF were significantly correlated (r = −0.469, *p* < 0.001; Table 4, Figure 5A). The run-down of the GRF and the peak CHF were also significantly correlated (r = −0.362, *p* < 0.016; Table 4, Figure 5B). Among the CHF variables, the PreMax and PreMin forces were significantly correlated with the peak CHF (r = 0.633, *p* < 0.001 and r = 0.332, *p* < 0.027, respectively; Table 4, Figure 5C,D). Among the GRF variables, only the run-down force was statistically significantly correlated with the peak CHF (r = −0.362, *p* < 0.016; Table 4, Figure 5B).

## 4. Discussion

This study aimed to analyze the forces generated by a therapist’s hands and feet over time during SM. We also analyzed the correlation between the time and force generated by the therapist’s hands and feet. To the best of our knowledge, this is the first study to simultaneously measure and analyze the force and time spent by the therapist’s hands and feet during SM training.

In this study, the run-up force of the feet appeared earliest in the temporal sequence of the SM, showing a downward incisural point characterized by a change from a decrease to an increase in force. This is related to the PreMax force characteristic of the hands, which follows the run-up of the feet. As described by Triano et al. [3] and Downie et al. [16], the maximum preload of the CH is localized by moving the target joint to its physiological limit, resulting in a quasi-static load, unlike the section of rapid weight movement and instantaneous dynamic load.

As the run-up force of the feet decreased, the maximum preload force of the hand increased, showing a negative correlation (r = −0.469, *p* < 0.001). This suggests that the therapist’s body weight was moved and supported by the hand, reducing the force of the feet and increasing that of the hands. The PreMax of the hand occurs later than the run-up of the feet; thus, it can be inferred that the force of the run-up of the feet influences the PreMax force of the hand but not vice versa. This suggests that the weight transfer to the hand occurs to localize the target joint and create a tension point. Judging from the start of the run-up and PreMax force changes, the feet were used as a driving force to create potential energy, which was subsequently used by the hands to lift the upper body. The PreMax forces in this study (288.26 ± 43.89 N) are similar to the maximum preload (240 ± 56 N) reported in a previous study by Herzog et al. [13], which applied SM to the SIJ.

In this study, the peak force of the feet is the point at which the force of the feet is maximized during the process of lifting the body, using the propulsion of the feet to create potential energy. This is characterized by an upward incisural point that changes from an increase to a decrease in force. The peak force of the feet in this study (617.48 ± 97.41 N) was similar to that reported in a previous study by Descarreaux et al. [17] for the experienced group (594 ± 29 N). The PreMin appears on the hand after the peak force of the feet is reached. The PreMin was found to be 118.67 ± 26.51 N and showed the characteristics of an upward incisural point that changes from a decrease to an increase in force. The peak force of the hand showed a positive correlation with both PreMax (r = 0.633, *p* < 0.001) and PreMin (r = 0.332, *p* < 0.027). The PreMax makes it easier to generate thrust by localizing the target joint [21]. The PreMin is the starting point of the thrust and the final preload that localizes the joint. Therefore, an increase in the localizing force of the target joint increases the peak force of the hand.

Many therapists agree that joint localization is essential for successful SM [14]. Elder et al. [20] reported that the thrust force was derived from the localization force of the target joint; however, this was not based on evidence. The present study provides important evidence that the joint localization force of the PreMax and PreMin induces and amplifies the thrust force. The hand PreMin is the point at which the preload force decreases after the PreMax. During this process, the force of the feet increases, lifting the body and generating potential energy. The hands appeared to use the PreMax force as a repulsive force to lift the upper body. Therefore, as the body is lifted and lowered, the change in the upward and downward movement of the body mass results in a reduced PreMin force of the hand, which is less effective for joint localization.

In this study, the run-down force of the feet was measured and found to be 139.98 ± 64.20 N, showing the characteristics of the downward incisural point, which changes from a decrease to an increase in force. Descarreaux et al. [17] reported that the vertical GRF decreases owing to the negative acceleration occurring when the body drops while the therapist moves the body downward, which is identical to the characteristics of the force that causes the run-down of the GR in this study. The peak force of the hand appeared after the run-down of the feet. The force shows a change from increasing to decreasing and is the point at which the greatest force applied from the hand to the target joint occurs, which is similar to the peak force reported by Downie et al. [16].

The measured variable of the feet with the peak force of the hand showed a negative correlation with the run-down force of the feet (r = −0.362, *p* < 0.016). According to O’Donnell et al. [14], many therapists believe that thrust is generated by dropping the body, and they teach this accordingly; however, no evidence was provided to support this assumption. While we agree with the concept of generating thrust through body drops, our study showed a negative correlation between the decreasing run-down force in the feet and the increasing peak force in the hands. This suggests that the therapist’s coordinated movement strategy of avoiding supporting the dropping weight with the therapist’s feet and ground and supporting the weight with the hands is an important factor in creating and amplifying the thrust force.

The peak force of the hands during SIJ SM was 544.08 ± 64.35 N. Herzog et al. [13] reported that the force magnitude and characteristics of the SM applied to the thoracic spine and the lumbar spine were similar. Forand et al. [23] reported that the peak force magnitude of the hand during the SM applied to the thoracic spine was between 200 and 800 N, similar to the force magnitude of this study. In a previous study from Downie et al. [16] concerning the same SIJ SM as this study, the peak force magnitude was 515 ± 123 N, similar to the results of this study. However, the average peak force of the CH in SIJ SM in this study was higher. This may be because the cervical, thoracic, and lumbar spines are composed of amphiarthrosis that allows movement; however, Vleeming et al. [24] reported that the surface of the SIJ, one of the major joints of the pelvis, is composed of bends and ridges; this joint has a high surface friction index and is with morphological closure connected by the strongest connective ligament in the human body. We believe that SM applied to the pelvis appropriately reflects a treatment strategy that considers arthrophysiological differences, in that SM should be applied with greater force than the corresponding joints of the hemi-joints, which allows limited movement.

This study identified the characteristics of the force of the feet and hands during SIJ SM during side postural positioning. We found that the force of the hands occurred after that of the feet. Changes in the force of the feet and hands were in opposition, and these changes occurred before the incisural point of the foot and the next incisural point of the hands. This suggests that the foot affects the hand movement in the nearest timeframe, but weight shifting has already begun to create a force in the next timeframe outside the nearest timeframe. These findings suggest that the SM of the SIJ in side postural positioning requires an understanding and mastery of all treatment procedures, timely weight shifting, effective transfer of force, and complex coordinated motor skills, in which all hand and foot movements to generate force are in an organic relationship.

This study has a few limitations. The SM was performed by a single therapist, which is characterized by the skill level of a single experimenter and requires limited interpretation of the results. Further experiments should be conducted with a larger number of participants. Although there was thigh contact between the therapist and participant to prepare the participant’s posture before the thrust, it cannot be concluded that force was not transmitted through this contact. Therefore, research on this should be conducted in the future. Future studies of force generation that include movements of the therapist’s upper body along with the feet and hands and movements such as hip and knee flexion are needed.

## 5. Conclusions

The results of this study on sacroiliac joint SM in a side posture revealed a sequential pattern. We found that the force of the contact hand occurred after that of the feet. Moreover, the contact hand forces (CHFs) were positively correlated with the CH preload maximum and minimum forces but negatively correlated with the ground reaction force (GRF) run-down. The insights gained from this study hold implications for the fields of teaching, motor learning, and safety within the context of spinal manipulation.

## Figures and Tables

**Figure 1 healthcare-11-03023-f001:**
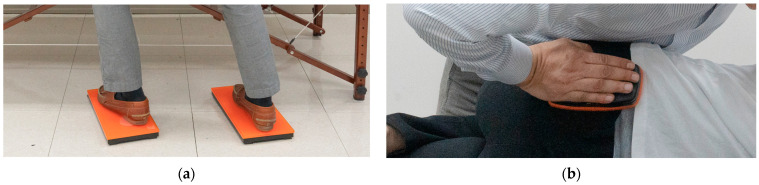
(**a**) Force plate between the experimenter’s feet and the ground; (**b**) Muscle controller (load cell) between the experimenter’s hand and the surface in contact with the skin.

**Figure 2 healthcare-11-03023-f002:**
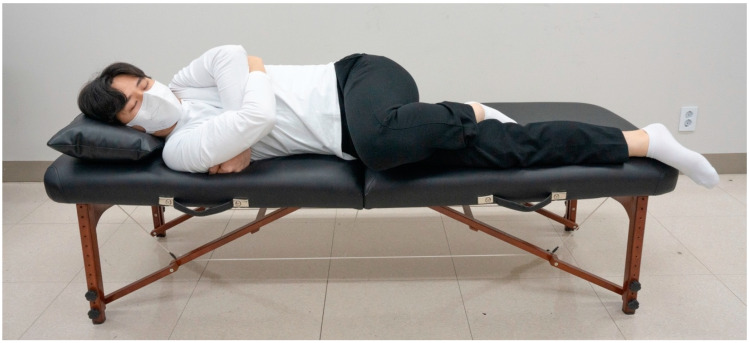
Side-lying position of the experiment subject.

**Figure 3 healthcare-11-03023-f003:**
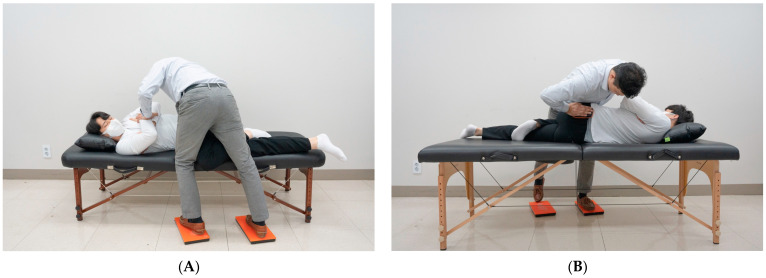
Therapist’s posture for sacroiliac joint manipulation. (**A**) Posterior view of the therapist; (**B**) Anterior view of the therapist.

**Figure 4 healthcare-11-03023-f004:**
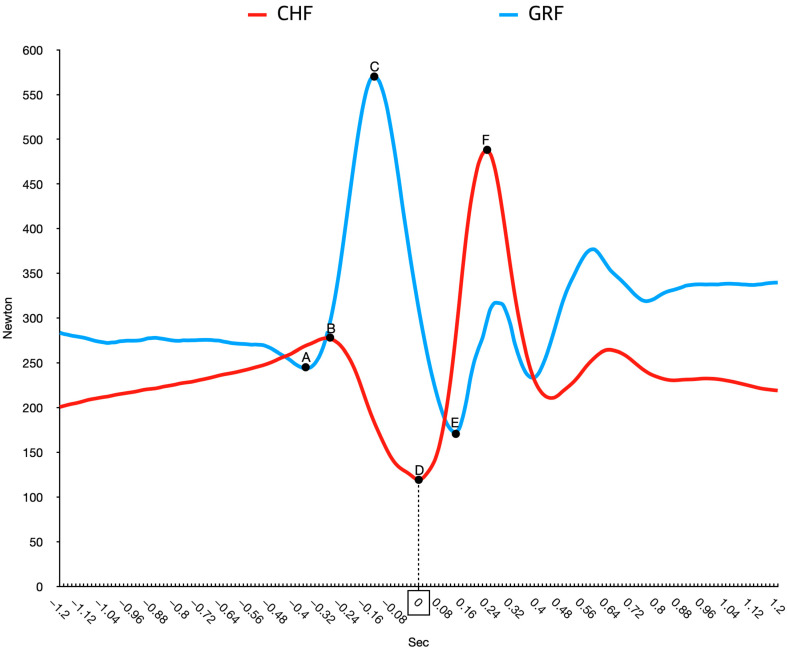
Force and time of the GR and CH during SM. (A) GRF RunUp: reduction in force before peak of the feet; (B) CHF PreMax: preload maximum force of the hands to localize the target joint; (C) GRF Peak: peak force of the feet; (D) CHF PreMin: preload minimum force of the hands is the final preload force that localizes the joint and is the start of the thrust; (E) GRF RunDown: the lowest force of the feet during SM; (F) CHF Peak: peak force applied to the target joint during SM.

**Figure 5 healthcare-11-03023-f005:**
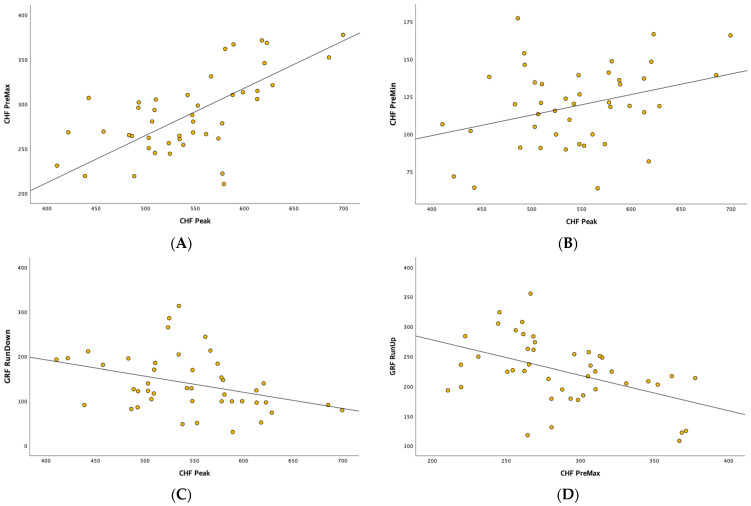
(**A**) Scatterplot of the CHF PreMax and CHF Peak; (**B**) scatterplot of the CHF PreMin and CHF Peak; (**C**) scatterplot of the GRF RunDown and CHF Peak; (**D**) scatterplot of the GRF RunUp and CHF PreMax.

**Table 1 healthcare-11-03023-t001:** General characteristics of the subjects.

	Healthy Adults (*n* = 44)
Age (years)	28.82 ± 3.28
Weight (kg)	70.32 ± 14.59
Height (cm)	170.77 ± 6.58

Values are mean ± SD.

**Table 2 healthcare-11-03023-t002:** Force and time of the GR and CH during SM (*n* = 44).

	Force Generated during SM (N)	Time Generated during SM (s)
GRF RunUp	225.28 ± 55.47	−0.34 ± 0.06
GRF Peak	617.48 ± 97.41	−0.15 ± 0.04
GRF RunDown	139.98 ± 97.41	0.11 ± 0.04
CHF PreMax	288.26 ± 43.89	−0.29 ± 0.06
CHF PreMin	118.67 ± 26.51	0.00 ± 0.00
CHF Peak	544.08 ± 64.35	0.21 ± 0.03

Values are mean ± SD; SM: spinal manipulation; GR: ground reaction; GRF: ground reaction force; CH: contact hand; CHF: contact hand force; N: Newton; s: seconds; PreMax: preload maximum; PreMin: preload minimum.

**Table 3 healthcare-11-03023-t003:** Force duration of the GR and CH during SM (*n* = 44).

	SM
GRF RunUp to Peak (s)	0.19 ± 0.05
GRF Peak to RunDown (s)	0.26 ± 0.04
GRF RunUp to RunDown (s)	0.46 ± 0.06
CHF PreMax to PreMin (s)	0.29 ± 0.06
CHF PreMin to Peak (s)	0.21 ± 0.03
CHF PreMax to Peak (s)	0.50 ± 0.05

Values are mean ± SD; SM: spinal manipulation; GR: ground reaction; GRF: ground reaction force; CH: contact hand; CHF: contact hand force; s: seconds; PreMax: preload maximum; PreMin: preload minimum.

**Table 4 healthcare-11-03023-t004:** Correlation between the GRF and CHF during SM.

	CHF PreMax	CHF PreMin	CHF Peak	GRF RunUp	GRF Peak	GRF RunDown
CHF PreMax	1					
CHF PreMin	0.296	1				
CHF Peak	0.633 ***	0.332 *	1			
GRF RunUp	−0.469 ***	−0.298 *	−0.172	1		
GRF Peak	−0.205	−0.269	−0.107	0.196	1	
GRF RunDown	−0.391 **	−0.347 *	−0.362 *	0.629 ***	0.187	1

* *p* < 0.05 ** *p* < 0.01 *** *p* < 0.001.

## Data Availability

Data are contained within the article.

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
