# Peer review of "Biomechanical Analysis of the Coordinated Movements of the Therapist’s Hands and Feet during Lumbopelvic Manipulation: A Preliminary Study"

_healthcare, 2023, doi:10.3390/healthcare11233023_

Round 1
Reviewer 1 Report
Comments and Suggestions for Authors
This is valuable manuscript about the hand and foot kinematics during the spinal manipulation. Here there are some comments and suggestions. Hope they are helpful.
Please use the lumbopelvic manipulation in the title instead of pelvic lumbar manipulation.
Please revise the key words according to the Mesh data base if it is possible.
What was the type of study?
What was the sample size estimation method?
Was the height of bed adjustable to the therapist height?
Please complete the title of figure “3” in the manuscript.
In the result part, some of finding are duplicated in the tables. Please report the summery of result according to the measured concepts.
Throughout the manuscript, there are several language mistakes. Therefore, I recommend a professional round of language editing before the paper is published. For example:
“measuring and analyzing the force and time of both the therapist’s hands and feet simultaneously in SM durning is a first for this study.” And “In this study, the RunUp force of the feet appeared earliest in the temporal sequence of SM, showing a downward incisural point characterized by a change from a decrease to an increase in force” and “The correlation of the measured variable of the feet with the peak force of the hand showed a negativecorrelationwiththerundownforceofthefee”
Please mention the most importance finding of your study in the conclusion part. You review the importance of topic in this section.
Comments on the Quality of English LanguageThroughout the manuscript, there are several language mistakes. Therefore, I recommend a professional round of language editing before the paper is published. For example:
“measuring and analyzing the force and time of both the therapist’s hands and feet simultaneously in SM durning is a first for this study.” And “In this study, the RunUp force of the feet appeared earliest in the temporal sequence of SM, showing a downward incisural point characterized by a change from a decrease to an increase in force” and “The correlation of the measured variable of the feet with the peak force of the hand showed a negativecorrelationwiththerundownforceofthefee”
Author Response
We express our deep gratitude to you and reviewers of “Healthcare” for taking their time to review our article. We have made corrections and clarifications in the manuscript after going over the reviewers’ comments. The corrections are included in the attached file. Please see the attachment.
Reviewer 1
Please use the lumbopelvic manipulation in the title instead of pelvic lumbar manipulation.
Response: Thank you for your kind suggestion. We followed your suggestion and modified the title to include the word ‘lumbopelvic’.
Please revise the key words according to the Mesh data base if it is possible.
Response: Thank you for the good suggestion. Following your suggestion, we tried to modify the keywords based on Mesh terms.
What was the type of study?
Response: The type of this study is a biomechanical observational study. This study aimed to observe and analyze the biomechanics of force transmission and generation during spinal manipulation performed by a single therapist on healthy subjects.
What was the sample size estimation method?
Response: The sample size was calculated through a pilot study. Through a pilot study, we examined the factors influencing specific patterns of hand and foot movements and their impact on the peak values of CHF. In the pilot study, we observed a Pearson correlation of R=0.637 between CHF PreMax and CHF Peak, as well as a correlation of R=0.42 between GRF's RunDown and CHF PreMax. Using G*Power software, we determined the required sample size for the correlation analysis. Setting the minimum correlation value at 0.42, alpha level at 0.05, and power at 0.8, we obtained a total sample size of 42 individuals. To account for the possibility of not achieving the experimental criterion of cavitation, we added 10%, resulting in a recruitment of 46 participants.
Was the height of bed adjustable to the therapist height?
Response: The height of the table can be manually adjusted according to the therapist's height, but in this study, it was adjusted to match the therapist's knee height according to suggestions in the literature. This content was additionally described in the research method.
Please complete the title of figure “3” in the manuscript.
Response: Thank you for your detailed suggestion. We have completed the title of figure 3.
In the result part, some of finding are duplicated in the tables. Please report the summery of result according to the measured concepts.
The times in Table 2 are a record of the points in time when force changes occurred in the hands and feet while SM started and progressed, and Table 3 records the intervals between each point of force generated in the hands and feet.
If you meant it to be redundant in some other sense, I hope you will kindly and generously point it out again.
Throughout the manuscript, there are several language mistakes. Therefore, I recommend a professional round of language editing before the paper is published. For example:
“measuring and analyzing the force and time of both the therapist’s hands and feet simultaneously in SM durning is a first for this study.” And “In this study, the RunUp force of the feet appeared earliest in the temporal sequence of SM, showing a downward incisural point characterized by a change from a decrease to an increase in force” and “The correlation of the measured variable of the feet with the peak force of the hand showed a negativecorrelationwiththerundownforceofthefee”
Please mention the most importance finding of your study in the conclusion part. You review the importance of topic in this section.
Response: Thank you for your detailed and kind review. In order to correct errors in English, we requested English proofreading from a professional agency.
Please mention the most importance finding of your study in the conclusion part. You review the importance of topic in this section.
Response: Thank you for your kind comments. Following your suggestion, we emphasized the results of the study and described the importance of the topic in the conclusion.

Reviewer 2 Report
Comments and Suggestions for Authors
Lines 216-7 describe very well the essence of this experiment. The authors conducted a good study on pelvic manipulation, with good knowledge of prior bibliography on this topic.
The limitations of this study is that one therapist was involved. It would have been more interesting if multiple therapists were involved. One of my questions is: how did the one therapist adjust the level of force application between subjects of different somatometric characteristics (height, weight, joint stiffness level, as directed by gender).
Perhaps the authors could focus on the:
Variability of upper & lower-limb forces application and coordination upon repeated administration of the technique on the same subject.
The variability noted between the healthy subjects the technique was applied to.
A detailed description of the technique involved could be added.
Also, please comment on whether there were forces transmitted to the SI joint through the therapist's thigh contact with the subjects.
Of course, the main issue with the manipulation is (as always) whether these well-co-ordinated forces have a direct biomechanical and neurophysiological effect to the joint they are directed to (specifically in the SI joint, for this study).
Finally, in my opinion, the technique (with practice) may not be very difficult to coordinate, as it is implied in the manuscript. However, a study between examiners could confirm this in the future.
Comments on the Quality of English LanguageGood quality
Author Response
We express our deep gratitude to you and reviewers of “Healthcare” for taking their time to review our article. We have made corrections and clarifications in the manuscript after going over the reviewers’ comments.
please see the attachment.

Round 2
Reviewer 1 Report
Comments and Suggestions for Authors
thanks a lot for correcting the manuscript.
Please mention the type of study in the method part.
Author Response
We truly appreciate your valuable reviews and feedback. Your insights are invaluable, and they help us improve our services. We look forward to your continued support and feedback in the future!
Please mention the type of study in the method part.
Response: Thank you for your kind review. The type of study is described in the method.